# Trichome Biomineralization and Soil Chemistry in Brassicaceae from Mediterranean Ultramafic and Calcareous Soils

**DOI:** 10.3390/plants10020377

**Published:** 2021-02-17

**Authors:** Tyler Hopewell, Federico Selvi, Hans-Jürgen Ensikat, Maximilian Weigend

**Affiliations:** 1Nees-Institut für Biodiversität der Pflanzen, Meckenheimer Allee 170, D-53115 Bonn, Germany; Ensikat@unitybox.de (H.-J.E.); mwei@uni-bonn.de (M.W.); 2Laboratori di Botanica, Dipartimento di Scienze Agrarie, Alimentari, Ambientali e Forestali, Università di Firenze, P.le Cascine 28, I-50144 Firenze, Italy; federico.selvi@unifi.it

**Keywords:** biomineralization, Brassicaceae, calcium carbonate, calcium phosphate, energy-dispersive X-ray spectroscopy, metallophytes, nickel hyperaccumulators, plant trichomes, scanning electron-microscopy

## Abstract

Trichome biomineralization is widespread in plants but detailed chemical patterns and a possible influence of soil chemistry are poorly known. We explored this issue by investigating trichome biomineralization in 36 species of Mediterranean Brassicaceae from ultramafic and calcareous soils. Our aims were to chemically characterize biomineralization of different taxa, including metallophytes, under natural conditions and to investigate whether divergent Ca, Mg, Si and P-levels in the soil are reflected in trichome biomineralization and whether the elevated heavy metal concentrations lead to their integration into the mineralized cell walls. Forty-two samples were collected in the wild while a total of 6 taxa were brought into cultivation and grown in ultramafic, calcareous and standard potting soils in order to investigate an effect of soil composition on biomineralization. The sampling included numerous known hyperaccumulators of Ni. EDX microanalysis showed CaCO_3_ to be the dominant biomineral, often associated with considerable proportions of Mg—independent of soil type and wild versus cultivated samples. Across 6 of the 9 genera studied, trichome tips were mineralized with calcium phosphate, in *Bornmuellera emarginata* the P to Ca-ratio was close to that of pure apatite-calcium phosphate (Ca_5_(PO_4_)_3_OH). A few samples also showed biomineralization with Si, either only at the trichome tips or all over the trichome. Additionally, we found traces of Mn co-localized with calcium phosphate in *Bornmuellera emarginata* and traces of Ni were detected in trichomes of the Ni-hyperaccumulator *Odontarrhena chalcidica*. Our data from wild and cultivated plants could not confirm any major effect of soil chemistry on the chemistry of trichome biominerals. Hyperaccumulation of Ni in the plants is not mirrored in high levels of Ni in the trichomes, nor do we find large amounts of Mn. A comparison based on plants from cultivation (normal, calcareous and serpentine soils, Mg:Ca-ratios ca 1:2 to 1:20) shows at best a very weak reflection of different Mg:Ca-ratios in the mineralized trichomes. The plants studied seem to be able to maintain highly conserved biomineralization patterns across a wide range of soil chemistries.

## 1. Introduction

Biomineralization plays a prominent role in the animal kingdom but it has received comparatively little attention in plants. It comes in essentially two different forms—the deposition of intracellular crystals or crystal complexes such as phytoliths and the mineral incrustations of plant surfaces, especially plant trichomes. Several hypotheses have been offered about the function of biomineralization in plants, including herbivore defense in the form of tissue rigidity and mechanical support, regulation of cytoplasmic calcium levels, detoxification of aluminum, heavy metals and/or oxalic acid, light gathering and scattering [1]. Plant trichome structure and chemical constituents often change according to function. This is especially true of trichomes exposed to light, herbivore damage, water stress, salinity or heavy metals [2]. Physical defense, widely considered the most important function of mineralized trichomes, is an example of this structural alteration, with effectiveness contingent upon trichome shape, size, density, placement and physical properties [2,3,4].

Silica, calcium carbonate (CaCO_3_) and calcium oxalate are the most widespread plant biominerals, with calcium carbonate being by far the most common biomineral, both in terms of the quantities produced and also its wide distribution in the plant kingdom [5,6]. Calcium carbonate and silica have long been known to play a prominent role in trichome biomineralization. Recent studies on a range of families, however, expanded our knowledge of plant biomineralization, demonstrating that calcium phosphate is also a widespread plant biomineral and that trichome biomineralization is both structurally and chemically much more complex than previously thought [4,7,8]. One of the plant groups shown to have trichomes mineralized with both calcium carbonate and calcium phosphate is Brassicaceae [8], a large and diverse family (ca. 328 genera and 3628 species; [9]) comprising numerous economically important representatives. Brassicaceae have a notoriously complex indument and the evolution of different trichome morphologies has been studied in detail [10]. Among the trichome types found in the family are malpighiaceous (or T-shaped) trichomes and stellate trichomes which should be more precisely characterized as radiate-stellate or peltate-stellate trichomes with dendritic branches [11]. The latter are characteristic of Brassicaceae Tribe Alysseae [10]. Brassicaceae trichomes are thus diverse in morphology and mineralization.

Members of the family Brassicaceae are found on gypsum [12], dolomite [13], limestone [14] and ultramafic soils. These soil types give rise to many specialized edaphic endemics in the Mediterranean basin [13,14,15]. Calcareous soils, derived from either chalk or limestone rocks, are noted for the presence of CaCO_3_, which can comprise up to 50% of soil content [16,17]. Serpentine soils are less common than calcareous soils and represent ca. 1% of the terrestrial surface with a patchy distribution [18]. Serpentine soils are quite unusual in their chemical composition: they contain >70% mafic (or “ferromagnesian”) material and less than 45% silica and are usually high in phytotoxic Ni, Co and Cr but very low in important plant nutrients such as N, P, K, Mo, B and Ca [19,20]. A distinctive feature of both serpentine and dolomite soil is the low Ca/Mg ratio, with a high concentration of Mg and significantly lower concentration of Ca compared to other soil types [15,19], albeit with considerable individual differences depending on the exact type of bedrock and different weathering regimes [21].

Serpentine and calcareous soils are often steep with a rocky granular texture that causes limited soil formation and sparse vegetation cover that lead to physically challenging conditions for plant growth due to soil erosion, low water retention capacity and heat stress [17,19,20,22]. These soil conditions give rise to a species-poor and specialized plant cover of highly adaptable generalists and edaphic specialists [17,23]. Serpentine endemics have evolved a number of adaptive traits similar to those of drought-adapted plants and cumulatively known as the “serpentine syndrome” [24]. The characteristic traits include reduced plant size, dense branching and smaller, thicker, more pubescent leaves than in non-serpentine counterparts [24,25]. Loew and May [26] were the first to study the influence of calcium and magnesium on the plant productivity of serpentine substrates and concluded that for optimal growth, the calcium to magnesium ratio must at least be equal. Physiologically, serpentine plants are able to react to Ca-deficiency by inverting the Ca:Mg ratio to values ≥1 within their tissues [24,27]. This is associated with one or more of the following adaptive mechanisms: tolerance of Ca deficiency and/or Mg toxicity, selectivity (ability to take up Ca in the presence of high concentrations of Mg) and luxury consumption of Mg (storage for later use; [22]). Serpentine endemics may also be able to cope with soil concentrations of Cr, Co and Ni well above toxicity thresholds for most other plants. Adaptations may involve either tolerance or avoidance, such as metal exclusion at the root level, compartmentalization in various plant organs and toxicity tolerance [22]. Some species are able to accumulate over 100 to 1000 times as much metal in their shoots and leaves as their non-hyperaccumulating relatives without displaying toxicity symptoms [22,28,29,30]. Minimum hyperaccumulation thresholds in plants are defined at 10000, 1000, 300 μg g^−1^ of leaf dry weight for Mn, Ni and Co, respectively [29]. Hyperaccumulation may be directed principally towards tolerance or disposal (sequestration), drought resistance, and/or herbivore or pathogen defense [31,32,33]. Sequestration is usually found in the epidermis, trichomes and cuticle of leaves, where toxic metals increase the defense function and are least likely to interfere with physiological processes such as photosynthesis [28,34,35]. Brassicaceae is by far the most important plant family in Western Eurasia with regards to hyperaccumulating species, with a total of 80 metal-accumulating species or nearly 12% of all known hyperaccumulators [29]. The Mediterranean basin is a major center of diversity for Ni-accumulating species, all of which belong to the two major tribes Alysseae and Noccaeae [36]. Brassicaceae are thus an important element of the vegetation on ultramafic soils—which are notably deficient in the elements crucial to biomineralization (Ca, P, Si) - and an important group of metal hyperaccumulators. At the same time, Brassicaceae trichome biomineralization has been shown to be more complex than previously assumed and a recent study, [34] demonstrated that at least one species (*Odontarrhena muralis* (Waldst. & Kit.) Endl.; syn. *Alyssum murale* (Waldst. & Kit.) can specifically accumulate manganese in the trichomes under controlled conditions. Brassicaceae, especially the metallophytes amongst them, are therefore an ideal group to study the connection between soil chemistry and trichome biomineralization.

In the present study we investigate patterns of trichome biomineralization for a range of different Brassicaceae from ultramafic and calcareous soils types. Included, a broad sample of narrowly endemic Ni-hyperaccumulators used to document patterns of trichome biomineralization in general and to specifically investigate whether divergent soil chemistry is reflected in trichome biomineralization in wild-collected material. Also, we brought different species into cultivation and exposed the same accessions to standard potting soil, ultramafic soil and calcareous soil to investigate whether and how, biomineralization in individual taxa reacts to divergent soil chemistry. We propose the following hypotheses:Soil chemistry is expected to be reflected in trichome biomineralization, especially in regard to Ca:Mg-ratios.Based on other plant groups, we expect the biomineralization patterns of Brassicaceae to be more diverse than so far reported.Hyperaccumulating species from serpentine soils are expected to incorporate elements such as Mn, Ni and Co into mineralized structures.

## 2. Results

### 2.1. Wild-Collected Samples

The most common trichome type found across our sampling is radiate-stellate trichomes with forked to dendritic branches (Figure 1A,B,E,F). In some species we find malpighiaceous (T-shaped) trichomes (Figure 1C,D). All trichomes have a very rough surface and compositional contrast signal shows that they are more or less completely mineralized. A closer view shows that even within the basic radiate-stellate trichomes, with forked to dendritic branches, there is considerable morphological diversity: some species have branches mostly forked once (Figure 1A), others mostly forked twice (Figure 1E,F) and sometimes there is higher-order branching (Figure 1B). Additionally, trichome branch surface is typically very rough, however, sometimes the branches are more or less smooth (Figure 2).

Across all 42 Brassicaceae samples obtained from wild populations (37 species from 9 genera), the trichomes are primarily mineralized with calcium carbonate, independent of soil type in the collection locality (Table 1). Calcium carbonate is the major biomineral in the stem and branches of the trichomes, only the very tip of the trichome branch shows some variability. Phosphorous is detected in the trichome tips of a total of eight samples from six genera (Table 1; *Alyssoides*, *Aurinia*, *Berteroa*, *Bornmuellera*, *Fibigia*, *Lepidotrichum*) indicating biomineralization with calcium phosphate or a mixture of calcium phosphate and calcium carbonate. Thus, *Fibigia clypeata* (Figure 2) has a striking concentration of phosphorous in the trichome tips, while *Bornmuellera emarginata* appears to have trichome tips consisting essentially of pure calcium phosphate (Ca to P peak ratio approximately 2:1, Figure 3).

All other 34 accessions show no P-signal but in 31 of the 42 samples (8 of 10 genera) there is a clear magnesium signal, with 25 samples showing a major magnesium peak. The presence of Mg in *Odontarrhena* varies considerably, ranging from completely absent (*O. decipiens)*, to abundant throughout (*O. stridii)*. As a rough proxy for different Ca:Mg ratios we compare spot peaks for Ca and Mg (Table 1). *Alyssoides utriculata* and 4 *Odontarrhena* species show an elevated concentration of Mg nearly throughout the entire trichome. We verify Mg and Ca signals along the trichome with energy-dispersive X-ray spectroscopy (EDX) line scans in *O. stridii*, demonstrating the near-uniform distribution of Ca and Mg (ratios ranging from 2.44:1 to 2.88:1; Figure 4).

A relative high proportion of magnesium is detected in most samples collected from ultramafic soils. *Berteroa incana* and *Lepidotrichum uechtrizianum*, collected on sandy soil, also reveal a high proportion of magnesium. There is a weak trend for samples from limestone to show much lower Mg-concentrations but the data as collected are not amenable to statistical analysis. Silicium is found in 4 of the samples (3 from *Odontarrhena* and 1 from *Bornmuellera*). In *Bornmuellera tymphaea*, Si is found predominantly in the tip of the trichome, where EDX spectra suggest that it replaces Ca almost entirely (Figure 5). In *O. chalcidica*, Si is found in the shaft and tip where it is co-localized with nickel (Figure 6).

### 2.2. Soil Analyses and Cultivated Plants

The amount of P present in all soil types is low or very low (Table 2). Magnesium concentrations are high across soil types but highest in serpentine soils (480–568 mg/kg). Similarly, Zn concentrations are high across soil types. The amount of Mn varies substantially between the soil types, with average concentrations (mg/kg) of 7.9–11, 36–42 and 54–63 for standard, calcareous and serpentine soils, respectively.

Serpentine soils are shown to have the lowest concentration of soluble calcium and the highest concentration of soluble magnesium for all species analyzed (Table 3). According to Ghasemi et al. [27], the Ca:Mg-ratios in the serpentine soils in our experiment (ranging from 1.6–2.2:1, Table 3) should be close to the developmental optimum for serpentine endemic Brassicaceae. The standard and calcareous soils have much higher Ca:Mg-ratios (ca. 5.9–19.5:1, Table 3).

The six accessions of *Bornmuellera* and *Odontarrhena* cultivated on the three different soil types show only marginal differences in their trichome mineralization between treatments. Trichomes are essentially mineralized with calcium carbonate – independent of soil Ca:Mg-ratios and the peak ratio Mg:Ca shows no striking patterns. Mg:Ca-ratios across the trichomes show broad overlap between treatments, there is no trend in the data with regards to manganese or calcium phosphate deposition in *B. emarginata*. The data reveals a common distribution of Ca throughout the base and shaft of the trichome. The element composition in the tips of the trichomes, however, shows more variation, being composed of CaCO_3_ + Mg or Ca_3_PO_4_ with some Mn (Table 4).

The type of mineralization is relatively uniform in *Odontarrhena*. CaCO_3_ and Mg are found throughout the trichome base, shaft and apex. *Bornmuellera emarginata* and *B tymphaea* are similar in their mineralization patterns, with CaCO_3_ in the base and shaft and abundant calcium phosphate at the tips. The ratio of P to Ca in the selected spots of the trichome tips is around 1:1.9 for *B. emarginata* and 1:3 for *B. tymphaea.* The ratio of P to Ca in analyzed spots of *B. emarginata* closely resemble that of pure apatite-calcium phosphate (Ca_5_(PO_4_)_3_OH), which is a typical component of vertebrate teeth and bones (Figure 3). Magnesium is found almost exclusively in the trichome tips of *B. emarginata* and in the trichome base, shaft and tip of all other species. Across the analyzed spots of *Odontarrhena*, ratios of Mg to Ca at the base (1:14), shaft (1:9.5) and apex (1:8.6) are determined. Interestingly, manganese is found in the trichome tips of *B. emarginata*, ranging from minimal traces to a Mn:Ca ratio of 1:7.5.

A rough analysis on the chemical composition ratios at the base, shaft and tip with ratios pooled based on soil type for all analyzed specimen reveals the ratio of Mg to Ca is the highest for all selected trichome locations in serpentine soil (Appendix A
Table A1) but a statistical analysis of this data does not show significant differences based on soil type. This indicates that there is a very weak, if any, correlation between the soil chemistry (amongst the soil types here used) and biomineralization. A rough statistical comparison of the spectra fails to detect a significant correlation between Mg-content and soil chemistry (Appendix A
Table A2).

## 3. Discussion

Our study on a range of different species of Brassicaceae from tribe Alysseae show comparatively uniform patterns of biomineralization, with universally mineralized trichomes and calcium carbonate as the dominant biomineral. Our first hypothesis is that the divergent soil chemistry of serpentine soils is reflected, for example, in the magnesium content of plants grown on this substrate. We find no conclusive evidence for this hypothesis from our data—plants appear to exert complete control over trichome mineralization, more or less independent of soil composition. Magnesium is a widespread minor component, apparently playing a subordinate role in wild plants collected from calcareous substrates and a slightly more prominent one in the trichomes of plants collected from serpentine soils. However, the data reported are neither quantitative nor do they permit a statistical analysis. Moreover, the different Ca:Mg ratios in the three different substrates used for plant cultivation do not appear to be correlated to the abundance of magnesium in mineralized trichomes. Within the range of Mg-contents covered by our experimental soil treatments (c. 8–63 mg/kg soil) and the different Ca:Mg ratios (2:1 to 20:1), there was no sufficiently high effect on the Mg:Ca-ratio in mineralized trichomes to be detectable by the techniques employed. The plants studied appear to be able to maintain the bulk composition of the biomineralized cell walls more or less identically across a wide range of soil parameters.

Our second hypothesis seeks to understand whether biomineralization patterns in Brassicaceae extend beyond the widely reported presence of calcium carbonate. This hypothesis is supported by our data, with the tips of the trichomes showing more variability in chemical composition than the base and shaft of the trichomes; here, we find high concentrations of phosphate across 6 of the 9 genera studied, sometimes with elevated contents of manganese and/or magnesium. Perhaps the most striking example of phosphate presence is found in the trichome tips of *Fibigia clypeatea*, where the P:Ca ratio is 1:2.1. The presence of phosphate in the trichomes studied is of particular interest given the lack of phosphate associated with serpentine and calcareous soils. In the soil used for cultivated samples, the amount of phosphate present is considered very low or low according to the classification system established by Finck [37]. It has been hypothesized that the alteration of chemical composition in trichome tips provides added hardness but experimental proof of this has not been provided thus far [3,4]. The accumulation of Mn (and Mg) may either be functional (detoxification, increased hardness) or a by-product of physiological processes. McNear and Kupper [38] found that manganese in Ni-hyperaccumulator *Odontarrhena muralis* is preferentially stored as Mn^2+^ in a complex with phosphate, in the basal compartment and rays of stellate trichomes on leaves and stems; at higher tissue concentrations, it was even found in the cell wall or apoplastic space of neighboring cells. In our case, we only looked at superficial biomineralization and confirmed the presence of manganese in the trichome tips. In Brassicaceae, amorphous calcium-phosphate has been recently documented in the trichome tips of the model organism *Arabidopsis thaliana* (L.) Heynh. [8]. The occurrence of amorphous calcium-phosphate in other lineages of Brassicaceae and particularly in Ni-hyperaccumulating species, however, is here demonstrated for the first time, underscoring that it is more widespread as a biomineral than previously documented. In trichome tips of wild collected *B. emarginata* the ratio of P to Ca was 1:2. The cultivated samples of this species showed similar results, with P present in the selected spots on the tips of the trichomes at a ratio of P to Ca of 1:1.9 for *B. emarginata* and 1:3 for *B. tymphaea*, closely approaching the ratio of pure apatite-calcium phosphate (Ca_5_(PO_4_)_3_OH). A similar composition was recently reported from the stinging hairs and scabrid-glochidiate trichomes of South American Loasaceae, where it is believed to replace silica to provide additional hardness [8,39,40].

In some species we found instances of biomineralization with silica, strikingly co-localized with nickel in the trichomes of *O. chalcidica*. Neither silica nor nickel have previously been reported in biominerals of Brassicaceae trichomes. Selvi and Bigazzi [41] found that Si was abundant in the leaf trichomes of Mediterranean Boraginaceae, often associated with Ca, where it was assumed to increase their mechanical stability and stiffness; no clear relationship with the soil type preferences of the 53 taxa analyzed and their element composition on the trichomes was found in Boraginaceae. Nickel is rarely reported as a biomineral but Broadhurst et al. [42], demonstrated the presence of nickel along with Mn in the basal compartment of trichomes on *O. muralis* grown in excess of Ni. The observation here represents the first documentation of nickel as part of biomineralized external cell walls.

Our third hypothesis predicts the presence of elements such as Mn and Ni in the trichomes of plants that are known hyperaccumulators of such elements. Our data finds no support for this hypothesis, with plants appearing to be able to exclude these elements from their mineralized trichomes. Despite the significant variation in the soluble concentration of manganese across soil types (7.9 mg/kg, 36 mg/kg and 54 mg/kg for standard, calcareous and serpentine soils, respectively), *B. emarginata* showed average Mn to Ca tip ratios of trace amounts in standard soil, 1:20 in serpentine soil and 1:13.8 in calcareous soil. Future research should focus on a possible function of Mn in trichomes and how *B. emarginata* appears to be able to accumulate similar concentrations of this element largely independently of soil Mn-levels. Identification of the molecular pathway responsible for the uptake and storage of Mn may provide an answer to this question. McNear and Kupper [38] found that the Ni-hyperaccumulator *O. muralis* grown at high Mn concentrations develops Mn-rich lesions around some leaf trichomes, in which large amounts of Mn^3+^ were found possibly for protecting cells from oxidative damage. Despite the elevated levels of soluble manganese and zinc in the serpentine soil used for cultivation [37], the lack of variation in trichome composition would suggest these metals do not significantly influence the mineralization of Brassicaceae trichomes.

Overall there is little evidence for a distinct effect of different soil types on the chemical composition of the trichomes. This is in stark contrast to the ability of many of the taxa here studied to hyperaccumulate elements such as nickel in their tissues [22,28,29]. The present study could, however, expand our understanding of trichome biomineralization in Brassicaceae: calcium carbonate is clearly the dominant biomineral but we could add another report of calcium phosphate and a first report of silica. Magnesium appears to be a widespread element in the calcified trichomes but it is unknown how it is integrated into distinct minerals. We could also add reports of traces of nickel and manganese in Ni-hyperaccumulating members of the family, in line with the elevated levels of both elements in the ultramafic soils they are specialized for. The present data add to the growing body of evidence indicating that plants exert an extraordinary degree of control over the exact type and localization of trichome biomineralization, inviting studies both in the functional significance of mineralized trichomes and the molecular mechanisms of the process of mineralization.

## 4. Materials and Methods

### 4.1. Plant Material

Plant material was collected mostly by the authors during field excursions in southern Europe and the Middle East, including sites with ultramafic outcrops in Albania, Greece and Italy (Appendix A
Table A3). We collected shoots and leaf samples from 42 accessions belonging to 37 species in 9 genera of tribe Alysseae DC. and 1 of tribe Arabideae (*Draba* L.), as defined by the established Alybase website by Španiel [43]. The genus *Leptoplax* O.E. Schulz, however, was included in *Bornmuellera* according to later phylogenetic and taxonomic evidence [44]. The sampling included wild collected material from different soils types, including sand, limestone and 26 of them came from serpentine soils. Twenty-four accessions correspond to known Ni-hyperaccumulators in the genera *Alyssoides* Medik., *Bornmuellera* Hausskn. (including *Leptoplax* O.E. Schulz) and *Odontarrhena* C.A. Mey. In addition, mature seeds were collected from 2 Ni-hyperaccumulator species of *Bornmuellera* from Greece and from 4 species of *Odontarrhena* (incl. 2 Ni-accumulators) from Greece and Italy, including the recently described endemic *O. stridii* [45]. These seeds were used to raise plants at the Botanische Gärten der Universität Bonn, Germany. The full list of taxa and accessions analyzed in the study is given in Table 1, with collection localities and voucher specimens in the *Herbarium Centrale Italicum* (FI, field collected samples).

### 4.2. Soil Analysis and Plant Cultivation

An analysis of soluble micronutrients in the calcareous, serpentine and standard soils used for cultivation of living material was performed before transplanting the seedlings. Soil was dried at 45 °C for four days and particulate matter greater than 2 mm was sorted out with a sieve. The determination of the soluble concentration of phosphorous was performed with a CAL solution (mixture of calcium acetate and calcium lactate), where phosphorous is extracted by shaking 5 g of soil for 90 min with 100 mL solution containing 0.1 M calcium acetate, 0.1 M calcium lactate and 0.3 M acetic acid (pH 4.1). Magnesium was determined with a CaCl_2_ solution (method from [46]) and iron, manganese and zinc were determined with a CAT solution (calcium chloride + Di-ethyl-triamine-penta-acetate (DTPA) (method from [47]). The determination of the cation exchange capacity (CEC) of calcium and magnesium in the different soils was performed using an NH_4_Cl extract and the resulting molality (mol/kg) concentrations were compared to determine the ratio of calcium to magnesium. The amount of soluble P, Mg, Mn and Zn, as well as pH in the soils, was classified according to [37].

Seeds obtained from the field were sown in sterilized standard TKS1 potting soil (pH-5.6; salt content-0.8 g/L; nitrogen-140 mg/L; phosphate-80 mg/L; potassium-190 mg/L; Floragard, Oldenburg, Germany). After 8 weeks, 5 plants were transplanted into 12 cm diameter pots with either standard TKS1 soil; natural serpentine soil, obtained from Tuscany, Italy (Lat. 43°19′ N; Long. 11°06′ E) and mixed 1:1 with TKS1 soil for potting; or calcareous soil with limestone gravel mixed 1:1 with TKS1 soil for potting. The effect of different soils on biomineralization was studied by sampling plants after growing them for >4 weeks in the respective soil types. The full list of cultivated accessions is given in Appendix A
Table A4, with collection localities and voucher specimens in the herbarium of the Nees-Institut für Biodiversität der Pflanzen (BONN, cultivated material).

### 4.3. Sample Preparation, SEM Observations and Processing of Element-Mapping Images

From both preserved field samples and cultivated (fresh) material from the garden, a 5–10 mm portion of the leaf was excised and mounted on its abaxial side for SEM analysis. The fresh leaf samples were fixed in a 70% ethanol +4% formaldehyde in water solution for 2 h, followed by washes of 85% ethanol, 99% ethanol, 1:1 ethanol acetone and 2 washes of 99% acetone. After dehydration, samples were critical point dried (CPD 40: Balzers, Liechtenstein). Samples were sputter-coated (SCD 040 Sputter-Coater: Balzers, Liechtenstein) with a thin layer (<30 nm) of palladium (Junker Edelmetalle, Waldbüttelbrunn, Germany) instead of gold, because the EDX spectrum of palladium does not interfere with the characteristic X-ray peaks of relevant elements (Si, P) relevant to biomineralization.

Topographic imaging and energy-dispersive X-ray (EDX) analyses were performed using a LEO 1450 SEM (Cambridge Instruments, Cambridge, UK) equipped with secondary electron (SE) and backscattered electron (BSE) detectors and an EDX analysis system with Link ISIS software (www.oxford-instruments.com). For evaluation of the mineral composition, the height of the element peaks was measured and compared. It was determined that analysis via peaks provided the most accurate analysis of element composition because several factors, such as surface morphology and surface layers of cutin lead to uncertainties. EDX analysis is not capable of measuring traces of elements; concentrations below ca. 0.2% would not be detected with this system. Images were recorded with a digital image acquisition system DISS 5 (Point Electronic, Halle, Germany). EDX mapping was performed and color images were generated by combining SE and BSE images. Image processing and color contrasting was carried out with standard image processing software (Paint Shop Pro 6, JASC, Eden Prairie, Minnesota, United States of America).

## Figures and Tables

**Figure 1 plants-10-00377-f001:**
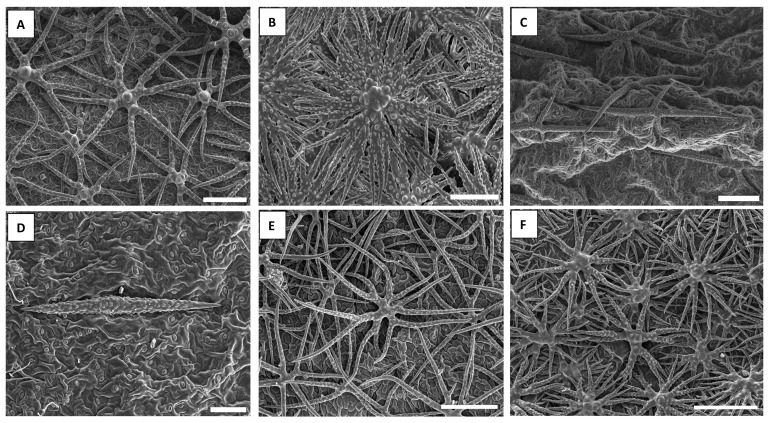
Secondary electron images showing the extent of trichome morphological diversity in Brassicaceae. (**A**). *Odontarrhena corymbosoidea* (*Selvi* & *Bettarini*, FI-055806, FI); many branched (radiate-stellate) morphology. (**B**). *Odontarrhena euboea* (*Cecchi* & *Selvi*, FI-058615, FI); complex higher order branching. (**C**). *Bornmuellera baldaccii* (*Selvi* & *Bettarini*, FI-055815, FI); forked malpighiaceous trichome. (**D**). *Bornmuellera emarginata* (*Selvi* & *Bettarini*, FI-055789, FI); stalkless T-shaped malpighiaceous trichome. (**E**). *Odontarrhena chalcidica* (*Selvi* & *Bettarini*, FI-055792, FI); unicellular stellate trichomes. (**F**). *Phylloloepidum cyclocarpum* (*Cecchi* & *Selvi*, Herb. FI-1362, FI); complex higher order branching. Scale bars: (**A**,**C**,**E**,**F**) = 200 µm; (**B**) = 90 µm; (**D**) = 100 µm.

**Figure 2 plants-10-00377-f002:**
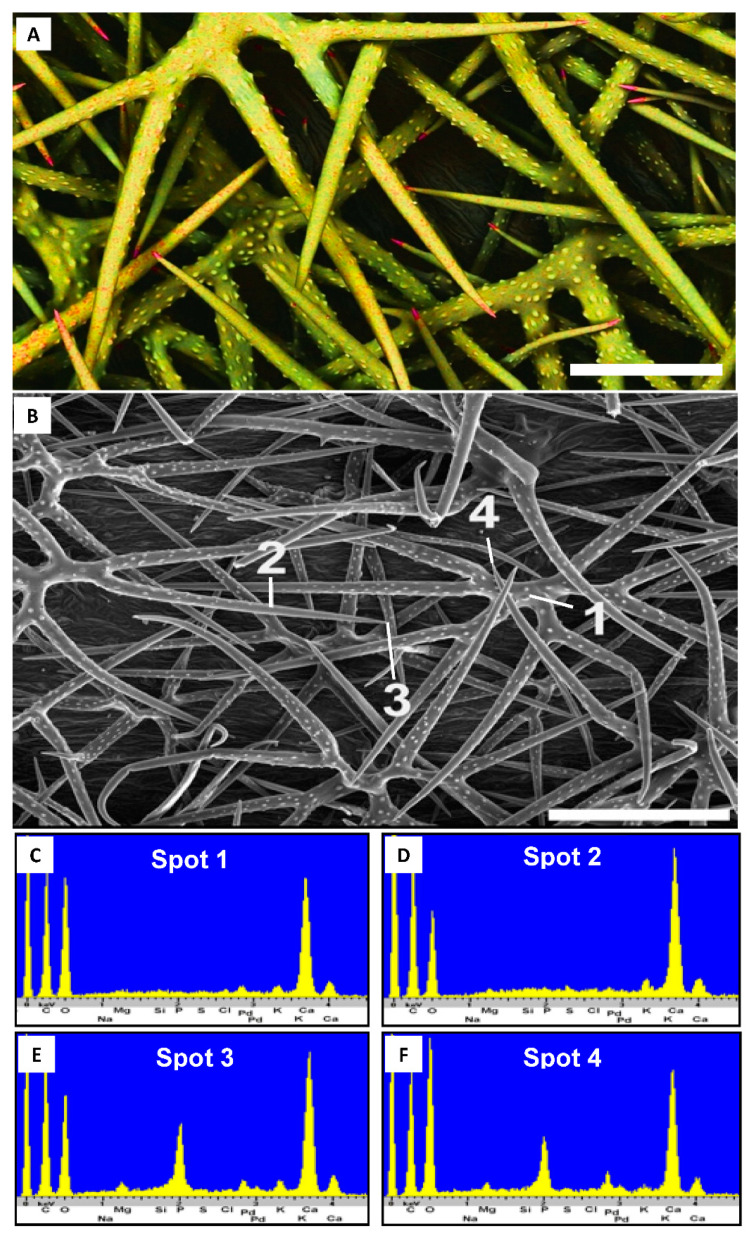
Trichome of *Fibigia clypeata* (*Selvi*, FI-058623, FI). (**A**). Secondary electron image depicting small tip regions containing phosphorus in high concentrations as calcium phosphate; depicted by red color. (**B**). Secondary electron image with analyzed spots associated with energy-dispersive X-ray spectroscopy spectra in (**C**) (base), (**D**) (shaft), (**E**) (tip) and (**F**) (tip). Scale bars: (**A**), 200 µm; (**B**), 100 µm.

**Figure 3 plants-10-00377-f003:**
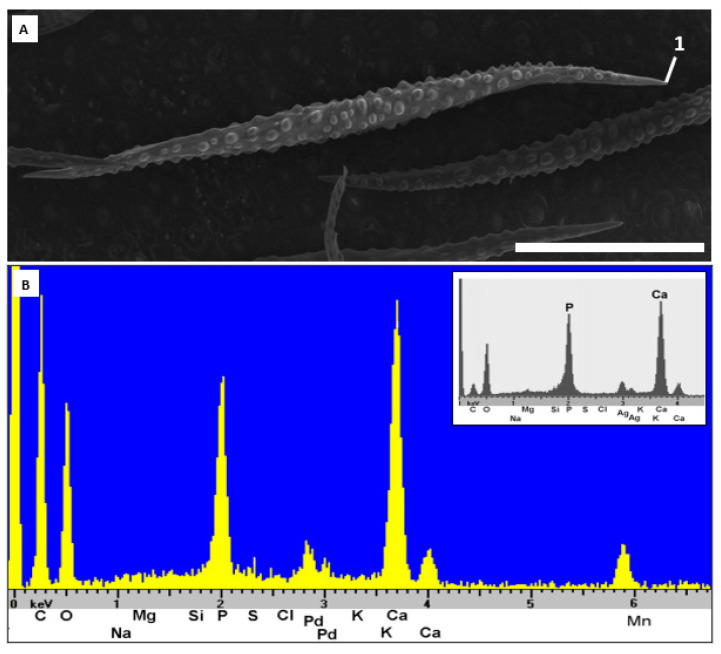
Cultivated *Bornmuellera emarginata* trichome (calcareous soil; accession 40470, BONN). (**A**). Secondary electron image of characteristic malpighiaceous trichome. (**B**). Energy-dispersive X-ray spectroscopy peaks at tip (spot 1) indicate a composition close to that of pure apatite-calcium phosphate (inset). Scale bar: 200 µm.

**Figure 4 plants-10-00377-f004:**
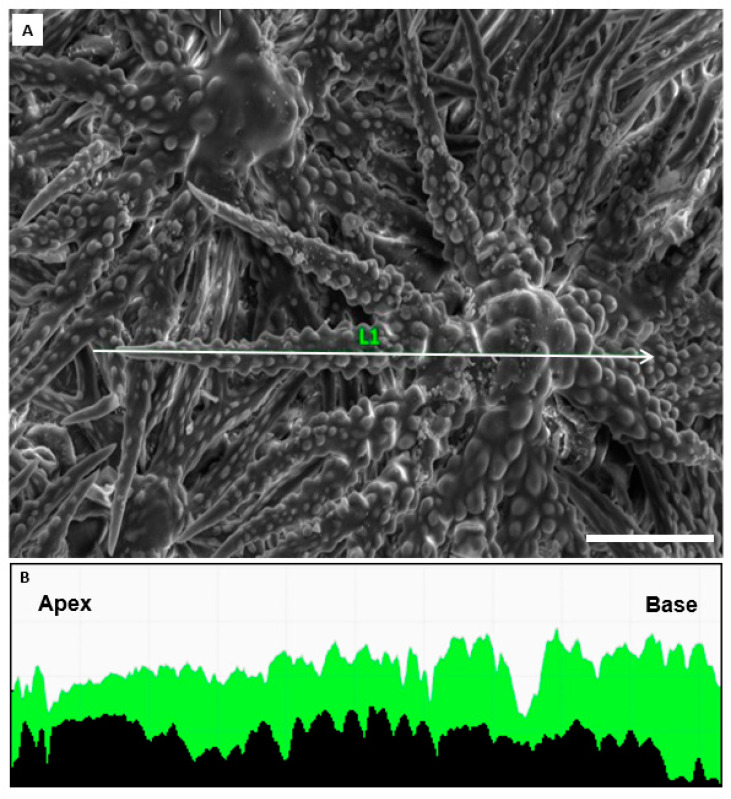
*Odontarrhena stridii* trichome analysis (collected from serpentine soil in Greece; Selvi & Bettarini, FI-055800, FI). (**A**). Secondary electron image of stellate trichome. (**B**). Energy-dispersive X-ray spectroscopy line scan results with the distribution of Ca (green) and Mg (black) illustrated from apex to base. The ratio of Mg to Ca for this selected branch averages to 1:2.88 in the apex, 1:2.44 in the shaft and 1:2.58 in the base. Scale bar: 50 µm.

**Figure 5 plants-10-00377-f005:**
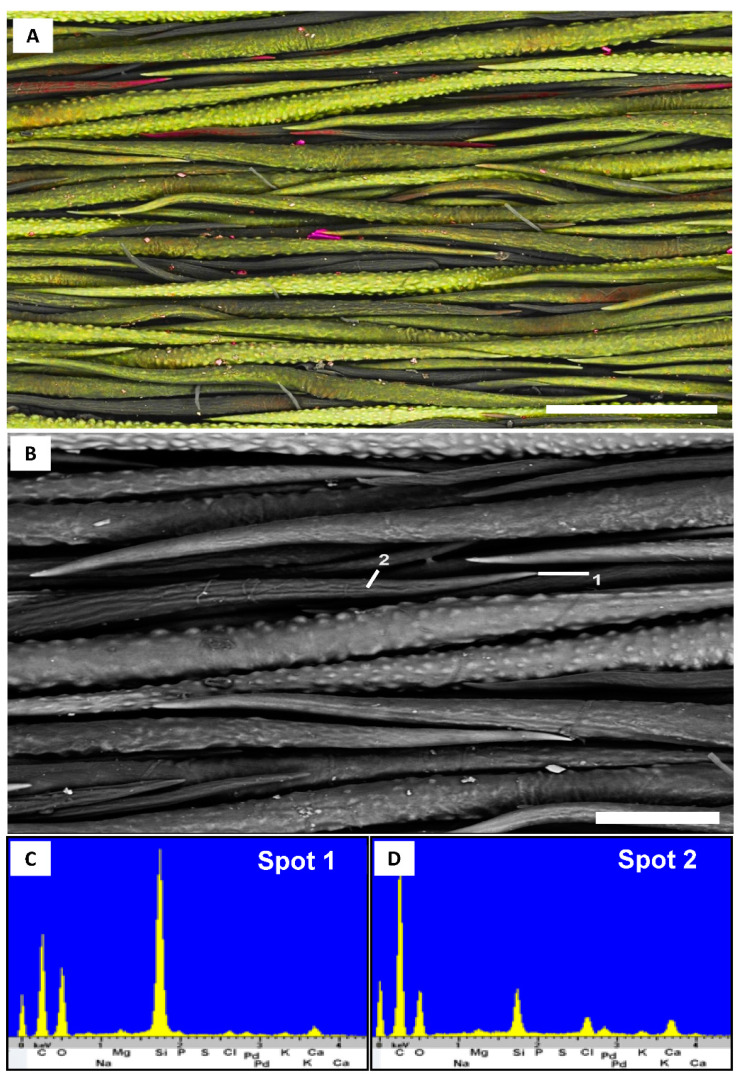
Silicium in *Bornmuellera tymphaea* (*Selvi* & *Bettarini*, FI-055791, FI). (**A**). Topographic (secondary electron) image combined with element mapping images of Si (red) and Ca (yellow) shows distribution of Si primarily in the tip but also in the body. (**B**). Secondary electron image with analyzed spots associated with energy-dispersive X-ray spectroscopy spectra labeled. (**C**). Energy-dispersive X-ray spectroscopy spectra of spot 1 (tip). (**D**). Energy-dispersive X-ray spectroscopy spectra of spot 2 (shaft). Energy-dispersive X-ray spectroscopy spectra depict the concentration of Si in relation to other elements present. Scale bars: (**A**), 200 µm; (**B**), 70 µm.

**Figure 6 plants-10-00377-f006:**
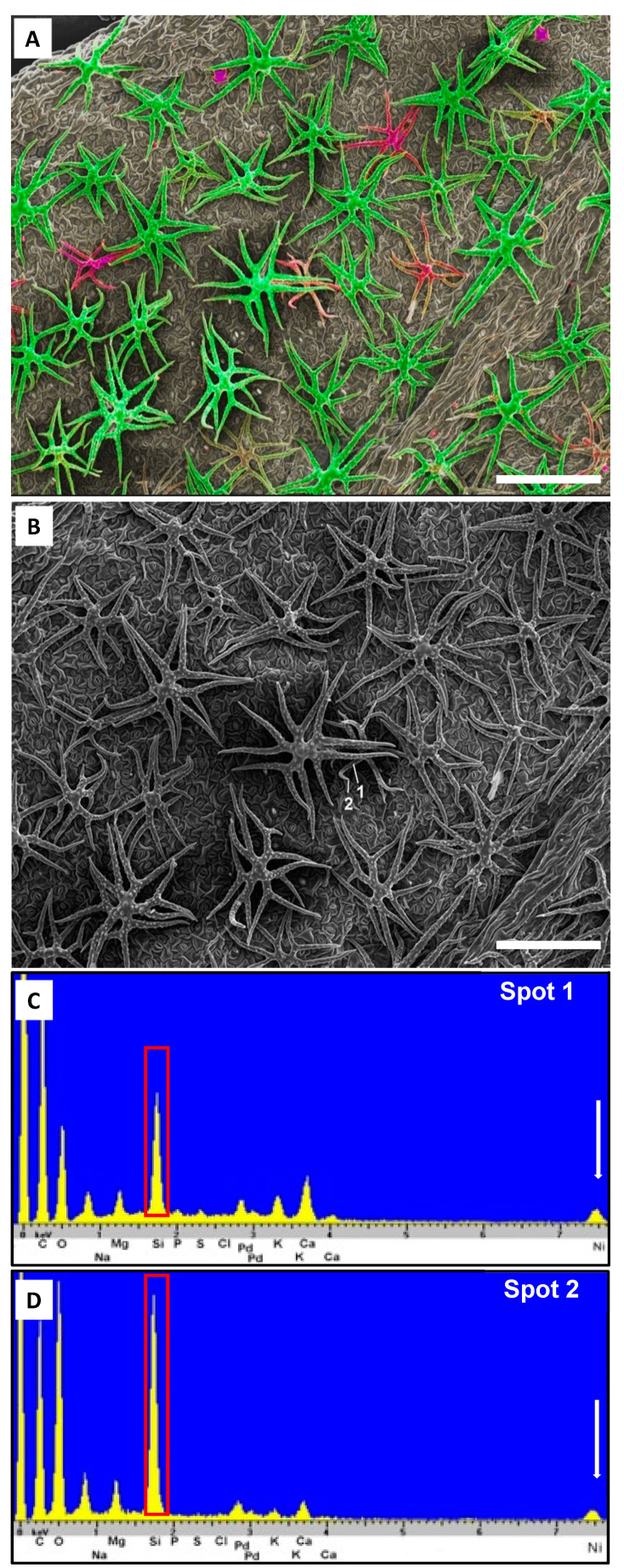
*Odontarrhena chalcidica* with Si and Ni present (*Cecchi* & *Selvi*, FI-050419, FI). (**A**). Secondary electron image is combined with element mapping of Ca (green) and Si (red) to illustrate mineralization pattern. (**B**). Secondary electron image with analyzed spots associated with energy-dispersive X-ray spectroscopy spectra in (**C**) (shaft) and (**D**) (tip), with Si (denoted by red box) and Ni (denoted by white arrow) presence shown. Scale bars: 200 µm.

**Table 1 plants-10-00377-t001:** List of examined taxa and accessions with soil type and elemental distribution of analyzed spots denoting the presence of Mg and P expressed as a ratio, and CaCO_3_ expressed as present or absent.

Specimen	Soil Type	Mg:Ca Base	Mg:Ca Shaft	Mg:Ca Tip	P:Ca Tip	CaCO_3_
*Alyssoides cretica* Medik.	Limestone	-	-	1:15	1:3.1	1,2
*Alyssoides utriculata* (L.) Medik. *	Serpentine	1:2.3	1:3.6	-	1:2.6	1,2
*Alyssum montanum* L. subsp. *montanum*	Limestone	-	-	-	-	1,2,3
*Aurinia saxatilis* (L.) Desv.	Limestone	-	-	-	1:4.2	1,2
*Berteroa incana* (L.) DC.	Sand	1:3.3	1:10.6	1:13	1:3.5	1,2
*Bornmuellera baldaccii* (Degen) Heywood *	Serpentine	-	-	-	-	1,2,3
*Bornmuellera emarginata* (Boiss.) Rešetnik *	Serpentine	-	-	1:13	1:1.9	1,2
*Bornmuellera emarginata* (Boiss.) Rešetnik *	Serpentine	-	-	-	1:6.9, 1:2.3	1,2
*Bornmuellera kiyakii* Aytaç & A. Aksoy *	Serpentine	1:7	1:18	-	-	1,2,3
*Bornmuellera tymphaea* Hausskn. *	Serpentine	1:35	1:12	1:35, 1:25	-	1,2,3
*Draba lasiocarpa* Rochel ex DC.	Serpentine	-	-	1:16	-	1,2,3
*Fibigia clypeata* Medik.	Limestone	-	-	1:16	1:2.1	1,2
*Lepidotrichum uechtrizianum* (Börnm.) Velen. & Börnm.	Sand	1:19	1:20	1:14	1:2	1,2
*Odontarrhena akamasica* (B.L. Burtt) Španiel, Al-Shehbaz, D.A.German & Marhold *	Serpentine	1:16	1:10.3	1:10.9, 1:7.8	-	1,2,3
*Odontarrhena albiflora* (F.K. Mey.) Španiel, Al-Shehbaz, D.A.German & Marhold	Limestone	1:14	1:14	1:10	-	1,2,3
*Odontarrhena argentea* (All.) Španiel, Al-Shehbaz, D.A.German & Marhold *	Serpentine	-	-	-	-	1,2,3
*Odontarrhena baldaccii* (Vierh. ex Nyár.) Španiel *	Serpentine	-	1:13	-	-	1,2,3
*Odontarrhena bertolonii* (Desv.) Jord. & Fourr. *	Serpentine	-	-	-	-	1,2,3
*Odontarrhena chalcidica* (Janka) Španiel, Al-Shehbaz, D.A.German & Marhold *	Serpentine	1:10	1:5.9	1:4.7	-	1,2,3
*Odontarrhena chalcidica* (Janka) Španiel, Al-Shehbaz, D.A.German & Marhold	Schist	1:23	1:19.5	1:18	-	1,2,3
*Odontarrhena chalcidica* (Janka) Španiel, Al-Shehbaz, D.A.German & Marhold	Schist	1:23	1:17	1:25	-	1,2,3
*Odontarrhena chalcidica* (Janka) Španiel, Al-Shehbaz, D.A.German & Marhold *	Serpentine	1:3.2, 1:4.5	1:5.8, 1:3.4, 1:2.2, 1:6.5, 1:3.4	1:2.5, 1:2.7, 1:2.8, 1:3.4	-	1,2,3
*Odontarrhena corsica* (Duby) Španiel, Al-Shehbaz, D.A.German & Marhold *	Serpentine	1:8	1:7.2	1:5.8	-	1,2,3
*Odontarrhena corymbosoidea* (Formánek) Španiel, Al-Shehbaz, D.A.German & Marhold	Limestone	1:7.4	1:8.6, 1:5.9	1:6.9, 1:5.4	-	1,2,3
*Odontarrhena decipiens* (Nyár.) L.Cecchi & Selvi *	Serpentine	-	-	-	-	1,2,3
*Odontarrhena euboea* (Halácsy) Španiel, Al-Shehbaz, D.A.German & Marhold *	Serpentine	-	1:13	1:8.9	-	1,2,3
*Odontarrhena fragillima* (Bald. & Balansa) Španiel, Al-Shehbaz, D.A.German & Marhold	Limestone	1:19	1:9.1	1:7, 1:12.6	-	1,2,3
*Odontarrhena heldreichii* (Hausskn.) Španiel, Al-Shehbaz, D.A.German & Marhold *	Serpentine	-	-	-	-	1,2,3
*Odontarrhena moravensis* (F.K.Mey.) L.Cecchi & Selvi *	Serpentine	-	-	1:9	-	1,2,3
*Odontarrhena muralis* (Waldst. & Kit.) Endl.	Granite	1:28	1:17, 1:15, 1:20	1:34	-	1,2,3
*Odontarrhena nebrodensis* (Tineo) L.Cecchi & Selvi subsp. nebrodensis	Limestone	-	-	1:7	-	1,2,3
*Odontarrhena peltarioidea* (Boiss.) Španiel, Al-Shehbaz, D.A.German & Marhold *	Serpentine	1:3.3	1:3.5	1:3	-	1,2,3
*Odontarrhena rigida* (Nyár.) L.Cecchi & Selvi *	Serpentine	-	-	-	-	1,2,3
*Odontarrhena robertiana* (Bernard ex Gren. & Godr.) Španiel, Al-Shehbaz, D.A.German & Marhold *	Serpentine	1:24	1:13	-	-	1,2,3
*Odontarrhena serpyllifolia* (Desf.) Jord. & Fourr.	Non-serpentine	-	1:19	-	-	1,2,3
*Odontarrhena serpyllifolia* (Desf.) Jord. & Fourr. *	Serpentine (culta)	-	-	-	-	1,2,3
*Odontarrhena sibirica* (Willd.) Španiel, Al-Shehbaz, D.A.German & Marhold	Serpentine	1:3.6	1:3.2	1:2.6	-	1,2,3
*Odontarrhena smolikana* (Nyár.) Španiel, Al-Shehbaz, D.A.German & Marhold subsp. *smolikana* *	Serpentine	1:23	1:21, 1:15	1:15, 1:16	-	1,2,3
*Odontarrhena smolikana* (Nyár.) Španiel, Al-Shehbaz, D.A.German & Marhold subsp. *glabra* (Nyár.) L.Cecchi & Selvi *	Serpentine	-	-	-	-	1,2,3
*Odontarrhena stridii* L.Cecchi, Španiel & Selvi *	Serpentine	1:4.6	1:2.9, 1:4.8, 1:2.6	1:7.1, 1:2.6	-	1,2,3
*Odontarrhena tavolarae* (Briq.) L.Cecchi & Selvi	Limestone	1:26	1:23, 1:44	1:21	-	1,2,3
*Phylloloepidum cyclocarpum* (Boiss.) L.Cecchi subsp. *pindicum* Hartvig	Limestone	1:15	1:13	1:18	-	1,2,3
* Ni-hyperaccumulating species.						
- denotes no Mg or P peak observed.						
1,2,3 in CaCO_3_ column denotes presence in base, shaft, and tip respectively.						

**Table 2 plants-10-00377-t002:** Analyses of standard, calcareous, and serpentine soil chemistry for cultivated specimen of Brassicaceae.

Specimen	Soil Type	mg P_2_O_5_/kg *	P	Mg (mg/kg) #	Mg	Fe (mg/L) °	Fe	Mn (mg/kg) °	Mn	Zn (mg/kg) °	Zn	pH (CaCl_2_)
*Bornmuellera emarginata*	Standard	15.8	A	312	E	5.8	trace	7.9	C	7.1	E	5.05
	Calcareous	39.8	B	118	D	0.84	trace	36	C	5.9	E	7.28
	Serpentine	15.1	A	568	E	9	trace	54	E	5.7	E	5.99
*Odontarrhena chalcidica*	Standard	27.5	A	320	E	5.7	trace	8	C	9.1	E	5.13
	Calcareous	33.3	A	97.8	D	0.73	trace	36	C	4.8	E	7.39
	Serpentine	14.9	A	512	E	9.9	trace	56	E	4.4	E	5.85
*Odontarrhena corymbosoidea*	Standard	45.0	B	301	E	5.5	trace	11	C	8.8	E	5.21
	Calcareous	35.2	B	87.7	C	0.75	trace	42	C	4.8	E	7.42
	Serpentine	17.9	A	480	E	9.8	trace	63	E	4.7	E	5.8
* CAL extraction solution	A: very low		A: < 34		A: < 30		normal:		pH dependent		A: <1	
° CAT extraction solution	B: low		B: 35 to 68		B: 30 to 40		10 to 50				C: 1 to 2.5	
# CaCl_2_ extraction solution	C: optimal		C: 69 to 137		C: 41 to 90						E: > 2.5	
	D: high		D: 138 to 275		D: 91 to 129							
	E: very high		E: < 275		E: > 120							
Class values based on Finck (1979).												

**Table 3 plants-10-00377-t003:** Concentration of soluble Ca and Mg in soil used during cultivation (mol/kg). Concentrations also expressed as a ratio of Ca to Mg.

Species	Soil Type	Ca (mol/kg)	Mg (mol/kg)	Ratio (Ca:Mg)
*Bornmuellera emarginata*	Standard	10.26	1.74	5.90:1
	Calcareous	9.44	0.56	16.86:1
	Serpentine	5.71	2.58	2.21:1
*Odontarrhena chalcidica*	Standard	11.27	1.95	5.77:1
	Calcareous	8.32	0.44	18.91:1
	Serpentine	5.00	2.80	1.79:1
*Odontarrhena corymbosoidea*	Standard	12.13	1.78	6.80:1
	Calcareous	8.92	0.46	19.52:1
	Serpentine	4.63	2.83	1.63:1

**Table 4 plants-10-00377-t004:** Elemental distribution of analyzed spots across the soil types of the species cultivated in Bonn. The presence of Mg, P, and Mn is expressed as a ratio. CaCO_3_ is expressed as present or absent.

Species	Soil Type	Mg:Ca Base	Mg:Ca Shaft	Mg:Ca Tip	P:Ca Tip	Mn:Ca Tip	CaCO_3_
*Bornmuellera emarginata*	Standard	-	1:14	1:13, 1:11	1:1.8, 1:1.7, 1:1.7, 1:10, 1:2.2	Trace	1,2
	Serpentine	1:10	1:11	1:8.6, 1:10	1:1.6, 1:1.6, 1:1.4	1:10, 1:20	1,2
	Calcareous	-	-	1:12	1:1.6, 1:2.2, 1:1.8, 1:1.6	1:11, 1:22, 1:7.5	1,2
*Bornmuellera tymphaea*	Standard	1:9	1:14	1:9, 1:20	1:10, 1:14, 1:1.7	Trace	1,2,3
	Serpentine	1:13, 1:16	1:13, 1:19	1:9, 1:15	1:4	Trace	1,2,3
	Calcareous	1:21, 1:18	1:22, 1:15	1:10, 1:6.6, 1:13	1:1.5, 1:2.5	-	1,2
*Odontarrhena bertolonii*	Standard	1:15	1:20, 1:15	-	-	-	1,2,3
	Serpentine	1:18	1:18	1:13	-	-	1,2,3
	Calcareous	1:9.5	1:5.7, 1:14	1:7, 1:10	-	-	1,2,3
*Odontarrhena chalcidica*	Standard	1:30, 1:14	1:16	1:10, 1:25	-	-	1,2,3
	Serpentine	1:15, 1:21, 1:25	1:10	1:6, 1:13, 1:19	-	-	1,2,3
	Calcareous	1:16	-	-	-	-	1,2,3
*Odontarrhena corymbosoidea*	Standard	1:10, 1:12, 1:23	1:8, 1:9, 1:9, 1:14	1:7, 1:10, 1:17, 1:10	-	-	1,2,3
	Serpentine	1:12, 1:12	1:9, 1:5	1:6, 1:5.5	-	-	1,2,3
	Calcareous	1:14	1:13, 1:11	1:11, 1:11	-	-	1,2,3
*Odontarrhena muralis*	Standard	1:13, 1:13, 1:15	1:11, 1:9, 1:11	1:10, 1:4	1:6	-	1,2,3
	Serpentine	1:10, 1:12	1:6, 1:6	1:9, 1:6	-	-	1,2,3
	Calcareous	1:12	1:7, 1:15	-	-	-	1,2,3
- denotes no Mg, P, or Mn peak observed.							
1,2,3 in CaCO_3_ column denotes presence in base, shaft, and tip respectively.							

## Data Availability

Not applicable.

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
