# Peer review of "Trichome Biomineralization and Soil Chemistry in Brassicaceae from Mediterranean Ultramafic and Calcareous Soils"

_plants, 2021, doi:10.3390/plants10020377_

Round 1

Reviewer 1 Report

This manuscript describes the study of the biomineralization in the trichome of Brassicaceae from Mediterranean ultramafic and calcareous soils. This interesting study finds that the Ca, Mg, Si, and P contents in trichome are independent of soil and dependent on species. However, there are many mistakes and scientifically inadequate writing in this manuscript.. The authors should rewrite carefully. Some examples are indicated below.

Line 128; “42 samples” is first appeared in the text. The authors should explain “42 samples” in this sentence.

Figure 4; The color blue is not found in this figure.

Table 2; A lot of “A, B, C, …” are in the footnote for this table. The footnote should be rewritten.

Appendix Table A1; The authors should describe the composition of the Base, Shaft, and Tip.

There are four Appendix Tables in this manuscript, but only three appendices are cited in the text.

Reviewer 2 Report

This is an interesting and important contribution to knowledge of functional aspects of metallophytic plant species, useful also as an experimental approach towards understanding of morphological and physiological adaptations of plants to metalliferous soils. The paper merits publication in Plants. I have only some minor points to mention, which need to be addressed before publication.

One problem is related to taxonomy of species. It is not clear what nomenclature system has been used. According to World Flora Online, Odontarrhena muralis is a synonym for Alyssum murale, and Odontarrhena chalcidica is yet another synonym of A. murale. It seems that probably also names of other taxa could be problematic. A reference to Al-Shehbaz seems to be not sufficient, as Odontarrhena is indicated as a synonym to Alyssum there.

In Abstract, indicate the aim before methodological secription. Do not start a sentence with a numeral (line 15).

In Introduction, the second paragraph is too long. It is advised to divide it in several logically separated parts.

It would be desirable to add geographical coordinates for sampling sites in Materials and methods.

Results need to be consistently described in the past tense.

Do not capitalize "manganese" (line 101).

Reviewer 3 Report

This manuscript does an excellent job describing the components of trichomes from different Brassica species. The research is very well presented, however,  I still don't know why was this work undertaken. What was the hypothesis? Was it expected that trichome composition would change depending on available ions in different soil types? Was the composition of trichomes expected to be dependent of the evolutionary lineage? Did the authors place any of the analyzed characters on the predicted Brassica lineage? 

A few interesting observations were made, but I want the authors to explain why they undertook this work. Even if the original hypothesis was not supported, I think this context is important. Please put this research in biological context for the reader.

I would like to know if the analyzed species are spread across the Brassica genus or are found in a monophyletic group within this genus.

The lines in figure 1 identifying different EDX spectra are too fine.

Why are 7 spots labelled in Figure 6, but only 2 EDX spectra showed?

Figure 6 shows trichomes either have high levels of Si or Ca deposits within. This observation isn't mentioned in the text. The only trichome with EDX spectra reveals a mixture of Si and Ca in it. Why isn't this discussed?

Reviewer 4 Report

Title.- I'm not sure it's very appropriate. Are the plants studied on calcareous soils metallophytes? I am not even convinced that the ultramafic soil species contemplated in the study can be considered metallophytes. In any case, if there are data to consider them hyperaccumulators, the references should be included in Appendix Table 3. After these considerations, I think a title such as Trichome biomineralization and soil chemistry in Brassicaceae from Mediterranean ultramafic and calcareous soils would be more appropriate.

page 2, lines 62-64.- The Brassicaceae are said to be among the most representative families of ultramafic soils and it is suggested that they are abundant in others. Indeed, that is so. They are abundant in gypsum (eg Musarella et al., 2018) and dolomite (eg Salmerón et al., 2018) or limestone (eg Tomovic et al., 2014) soils, but I think some references should be added to support this idea.

p.2-3, lines 64-93.- Much is said here about ultramafic soils and their restrictions and characteristics, but nothing is said about calcareous ones. It is a curious asymmetry considering the title of the manuscript. I think the manuscript requires adding something about the calcareous soils. Some of their characteristics have a lot to do with the chemical analysis of trichomes. For example, calcareous soils are very poor in P and Ca plays a key role in sequestering this element (e.g. Ghoneim et al., 2020).

p. 3, lines 93-94. The family with the most hyperaccumulator species is not the Brassicacaceae, but Phyllantaceae. At least according to http://hyperaccumulators.smi.uq.edu.au/collection/ Crucifers rank second

p.3, lines 94-96.- It is also inaccurate that the Mediterranean is the main center of Ni-accumulating species. http://hyperaccumulators.smi.uq.edu.au/collection/. New Caledonia, Cuba or SE Asia have more species recognized as hyperaccumulators.

In table 1 I think it would be very interesting to know the types of soils in which the different species were collected. Although those characteristics are in Appendix table 3 they should be listed in table 1.

p. 9, lines 147-148.- What is said here about the high proportion of Mg in ultramafic and sand soils has caught my attention. Could this have to do with the soil texture – Do they come from dunes?

Table 2.- I am not very clear what exactly a “standard” soil is. Of course with a pH of 5, I think it is a quite acidic soil, even more so than ultramafic ones. In my opinion, it would be necessary to detail more precisely what this type of soils consist of in the M&M.

General comments

I do not understand why several of the functions attributed to trichomes in plants have not been mentioned in the discussion (e.g. Karabourniotis et al_2019; Bickford, 2016; Holmes & Keiller_2002). I think that the article would gain a lot if the discussion on other aspects such as reflectance or the water economy were expanded. In this way the discussion could be separated from a section dedicated to the conclusions. In my view, the results obtained would allow us to do this. The discussion seems rather hasty, and it is not easy to distinguish the results of the field study from those carried out under growing conditions.

I was very surprised that Odontarrhena tavolarae is a hyper-accumulating species. ¿On limestone?

Another aspect that perhaps needs to be emphasized a bit more is the presence of P in trichomes. P is not an abundant element in this type of soils, so its presence in trichomes is very striking.

Round 2

Reviewer 1 Report

The manuscript is not revised enough to be accepted for its publication. Thus I think this manuscript is not suitable to be published in Plants.
